# Regulation and Response Mechanism of Acute Low-Salinity Stress during Larval Stages in *Macrobrachium rosenbergii* Based on Multi-Omics Analysis

**DOI:** 10.3390/ijms25126809

**Published:** 2024-06-20

**Authors:** Xilian Li, Binpeng Xu, Peijing Shen, Haihua Cheng, Yunpeng Fan, Qiang Gao

**Affiliations:** Agriculture Ministry Key Laboratory of Healthy Freshwater Aquaculture, Key Laboratory of Fish Health and Nutrition of Zhejiang Province, Zhejiang Institute of Freshwater Fisheries, Huzhou 313000, China; lixilian@126.com (X.L.); scoffild@126.com (B.X.); shenpeijingzj@hotmail.com (P.S.); m130101022@163.com (H.C.); yunpengfan1994@163.com (Y.F.)

**Keywords:** low-salinity stress, omics, *Macrobrachium rosenbergii*, environment, acute stress

## Abstract

*Macrobrachium rosenbergii* is an essential species for freshwater economic aquaculture in China, but in the larval process, their salinity requirement is high, which leads to salinity stress in the water. In order to elucidate the mechanisms regulating the response of *M. rosenbergii* to acute low-salinity exposure, we conducted a comprehensive study of the response of *M. rosenbergii* exposed to different salinities’ (0‰, 6‰, and 12‰) data for 120 h. The activities of catalase, superoxide dismutase, and glutathione peroxidase were found to be significantly inhibited in the hepatopancreas and muscle following low-salinity exposure, resulting in oxidative damage and immune deficits in *M. rosenbergii*. Differential gene enrichment in transcriptomics indicated that low-salinity stress induced metabolic differences and immune and inflammatory dysfunction in *M. rosenbergii*. The differential expressions of *MIH*, *JHEH*, and *EcR* genes indicated the inhibition of growth, development, and molting ability of *M. rosenbergii*. At the proteomic level, low salinity induced metabolic differences and affected biological and cellular regulation, as well as the immune response. Tyramine, trans-1,2-Cyclohexanediol, sorbitol, acetylcholine chloride, and chloroquine were screened by metabolomics as differential metabolic markers. In addition, combined multi-omics analysis revealed that metabolite chloroquine was highly correlated with low-salt stress.

## 1. Introduction

Salinity in natural environments directly or indirectly affects growth, development, reproduction, behavior, and distribution of aquatic organisms and is a key factor in regulating and maintaining osmotic pressure inside and outside the organism [1]. The body of aquatic animals requires a timely response to salt imbalances in water, or salinity acclimation, which often involves complex molecular mechanisms [2].

*Macrobrachium rosenbergii* is a large freshwater prawn with fast growth, a short culture cycle, and high nutritional value. In the high-density intensive aquaculture environments, aquatic animals are frequently affected by various environmental factors present in living environment thus prolonging the production cycle and decreasing economic benefits [3,4]. Salinity is a fundamental aquatic environmental factor affecting crustacean aquaculture, including food conversion efficiency, metabolism, growth, and development of crustaceans, as well as other life activities [5,6]. High requirement for salinity in the larval period of *M. rosenbergii* leads to a high risk of salinity stress in the aquaculture water [7].

Crustaceans generally have an ideal salinity range for optimal growth and development, within which their feeding, larval development, and reproduction rates are the highest [8,9]. If the salinity fluctuates too much, beyond the salinity adaptation range of crustaceans, it is not favorable for their growth [10,11]. Research has found that the growth, development, and immune system of Rottweiler shrimp are adversely affected to some extent by salinities of below 12‰ [12]. Changes in salinity can affect the osmotic pressure of crustaceans’ bodies, which can impact their energy levels and ability to carry out normal life activities. Once a crustacean has adapted to a particular salinity environment, it no longer needs to expend energy to regulate its osmotic pressure, allowing for more energy to be allocated towards growth and development [13,14]. The vanabin prawn can survive in a salinity range of 0.5–35‰, with the fastest growth occurring at a salinity of 20‰; however, growth is slow when the salinity of the culture water is below 15‰ [15]. Crayfish are capable of surviving and growing in water with salinities below 14 ppt and are highly adaptable to salinities below 10‰ [16]. It was found that the survival rate of the Japanese sac shrimp decreases significantly with decreasing salinity [17]. The salinity safety value for crayfish with a body weight of (15.0 ± 2.0) g and juvenile crayfish with a body weight of (2.0 ± 0.1) g were 6‰ and 6.23‰, respectively, with a higher survival rate, faster growth and lower feed coefficient at salinities ranging from 0 to 6‰ [18].

Salinity changes have a significant impact on the growth, development, and osmotic adjustment of crustaceans, but also lead to accelerated physiological metabolism and energy consumption rates. Long-term exposure to salinity stimulation can result in disorders in immune and antioxidant capacity of organisms. [19]. Numerous reports have been published on the impact of salinity on the immune defense and physiological and biochemical indices in crustaceans. Research has found that in intertidal crabs (*Macrophthalmus japonicus*) exposed to high osmotic environments, heat-excited proteins were involved in cellular homeostasis and immune defense in response to high exposure to high-salt environments [20]. A study on the impact of salinity on the immune defense of the Atlantic blue crab (*Callinectes sapidus*) found that a decrease in salinity resulted in increased blood cell aggregation and bacterial accumulation in the gill tissue of *Callinectes sapidus* [21]. In conclusion, crustaceans counteract changing salinity environments through the regulation of immune function.

The intricacy and diversity of biological processes, coupled with the intricate regulation of gene expression, create within individual histological studies a significant bottleneck because the conclusions drawn from single-omics studies are often incomplete [22,23]. The application of multi-omics techniques, which combine two or more genomics approaches, such as genome, transcriptome, proteome and metabolome, is a common practice in the field of systematic biological studies [24]. A comprehensive multi-omics analysis of a vast quantity of data from a multitude of biomolecules is conducted, encompassing normalization, comparative and correlation analyses to explore data relationships between molecules at different levels [25]. Simultaneously, the combination of Kyoto Encyclopedia of Genes and Genomes (KEGG) pathway enrichment, molecular interactions and other bio-functional analyses allows for a systematic and comprehensive analysis of the functions and regulatory mechanisms of biomolecules.

*M. rosenbergii*, a freshwater prawn, exhibits a low or even zero salinity requirement in its adult stage, while it has a higher salinity requirement in its reproductive and juvenile stages. Consequently, the greater salinity demand will result in increased salinity stress in the water during aquaculture. In this study, transcriptomics, proteomics and metabolomics were combined to jointly reveal the mechanism of response to low-salinity treatments on the larvae of *M. rosenbergii*. Studying the low-salt tolerance ability of *M. rosenbergii* can provide important theoretical support for its low-salinity nursery and acclimation, as well as a scientific basis for solving the problems in the development of the *M. rosenbergii* industry, which is of significant academic and application promotional value.

## 2. Results

### 2.1. Changes in Biochemical Parameters

The catalase (CAT), glutathione peroxidase (GSH-Px), and superoxide dismutase (SOD) activities in each group were analyzed to evaluate the immune response in *M. rosenbergii* under different salinity. In the hepatopancreas, significant alterations in the CAT, GSH-Px, and SOD activities were observed in low-salt-stressed *M. rosenbergii*, with the expression levels of the aforementioned biochemical indices being demonstrated as diminished in the SAL-0 and SAL-6 groups relative to the SAL-12 group. Moreover, except CAT, the GSH-Px and SOD activities exhibited a significant decline in the SAL-0 group in comparison to the SAL-6 group. In muscle, the expression levels of GSH-Px, and SOD enzymes activities in SAL-12 showed similar results. Low salinity caused an obvious decrease in GSH-Px, and SOD activities in *M. rosenbergii*. However, no significant differences were observed in CAT activity between the different groups (Figure 1). The results of this study indicate that low-salinity stress results in oxidative damage and impairs auto non-specific immunity in *M. rosenbergii*.

### 2.2. Transcriptome Analysis of Differentially Expressed Genes (DEGs)

Based on the Fragments Per Kilobase of exon model per Million mapped fragment (FPKM) values, the gene expression levels under different levels of salinity were calculated. A total of 739 DEGs were identified in the comparison between the SAL-0 and SAL-6 groups, comprising 376 upregulated genes and 363 downregulated genes (Figure 2A). In the comparison between the SAL-0 and SAL-12 groups, the significant expression of 2615 genes were observed, with 1567 genes being upregulated and 1048 genes being downregulated (Figure 2B). In the comparison between the SAL-6 and SAL-12 groups, the transcription of 2209 genes were significantly altered, with 1372 genes being upregulated and 837 genes being downregulated (Figure 2C). The coexistence of 14 DEGs in three comparison groups is shown in the Venn diagram (Figure 2D).

Biological function of DEGs determined by Gene Ontology (GO) enrichment analysis. Differentially expressed genes are mainly assigned to three major functional groups, namely cellular components, molecular functions, and biological processes. The largest subclasses of differences between the SAL-0 and SAL-6 groups were the toll-like receptor 1 and 2 proteins, structural molecule activity, and stratum corneum structural components, and sodium ion transport (Figure 3A). For the SAL-0 group vs. the SAL-12 group, the DEGs were mainly enriched in hematopoietic cell lineage, sister chromatid segregation, organelle fission, nuclear division, nuclear chromosome segregation, and mitotic sister chromatid segregation (Figure 3B). While comparing the SAL-6 and SAL-12 groups, protein kinase C inhibitor activity, structural components of the ocular lens, modulation of endothelial cell chemotaxis, response to interleukin-11, butyrate and modulation of cellular response to heat were measured (Figure 3C). The results of the official classification and KEGG annotation indicated that low-salinity stress led to the identification of *M. rosenbergii* DEGs enriched in specific biological pathways related to metabolism, organic systems and cellular processes, including the PI3K–Akt signaling pathway, NOD-like receptor signaling pathway, p53 signaling pathway, antigen processing and presentation, fatty acid degradation, lipid and atherosclerosis, glutathione metabolism, linoleic acid metabolism, pyrimidine metabolism, arachidonic acid metabolism, etc. (Figure 3D–F). This may suggest that *M. rosenbergii* induces immune and inflammatory dysfunction in the body by affecting metabolic pathways under low-salinity stress.

### 2.3. Expression Validation of Selected DEGs

Subsequently, the reliability of the transcriptome data was confirmed by qRT-PCR validation of the mRNA expression of the selected genes. The mRNA expression level of the Molt-inhibiting hormone (*MIH*), Retinoid X receptor (*RXR*), Juvenile hormone epoxide hydrolase (*JHEH*), Ecdysteroid receptor (*EcR*), Cysteinyl aspartate specific proteinase 3 (*Caspase 3*), Cysteinyl aspartate specific proteinase 8 (*Caspase 8*), cytochrome c (*Cyt-c*), tumor suppressor gene (*P53*), nuclear factor kappa-B (*NF-κB*), and B cell lymphoma 2 ovarian killer (*Bok B*) were analyzed. The qRT-PCR results were consistent with the differential expression results from RNA-Seq analysis. A significant downregulation of MIH mRNA expression levels was observed in the salinity exposure groups (0‰ and 6‰ salinity) compared to the control group (SAL-12). However, the mRNA expression levels of *JHEH*, *EcR*, *Caspase 3*, and *Caspase 8* in SAL-12 were found to be significantly downregulated in comparison to SAL-0 and SAL-6 (Figure 4).

### 2.4. Proteomic Analysis

The *M. rosenbergii* hepatopancreas under different salinities were further analyzed using proteomic analysis. As shown in Figure 5A, the number of the total spectrum is 643,549, while the matched spectrum is 108,013; the number of the total peptides is 18,155, and unique peptides is 17,737; the number of the identified proteins is 2662, and 2603 proteins is quantified. In order to analyze the repeatability of protein identification between the samples within the group, the overlap of samples within the group was analyzed using the Venn diagram. Figure 5B illustrates the co-occurrence of 2209 proteins identified in the three comparison groups. In order to determine the proteins with different expression levels between the different groups, the experimental data were further analyzed. Differentially expressed proteins (DEPs) were identified at *p* < 0.05 and FC values > 2 or <0.5. As shown in Figure 5C–F, the expression of 31 and 28 DEPs was upregulated, while that of 33 and 40 DEPs was downregulated in SAL-0 compared to SAL-6 and SAL-12, respectively. In the SAL-6 group, the expression of 26 DEPs was upregulated, while that of 25 DEPs was downregulated compared to the SAL-12 group.

The DEPs were subjected to further analysis using GO and KEGG. In the SAL-0 and SAL-6 groups (Figure 6A), DEPs are significantly enriched in the following biological processes: chitin metabolic, glucosamine-containing compound metabolic, aminoglycan metabolic, amino sugar metabolic, macromolecule diacylation processes. In terms of molecular function, the DEPs are mainly enriched in transferase activity, transferring pentosyl groups, protein deacetylase activity, large conductance calcium-activated potassium channel activity, deacetylase activity, and macromolecule transmembrane transporter activity. The cellular components mainly include pigment granules, the dense body, and melanosomes. In the SAL-0 and SAL-12 groups (Figure 6B), the DEPs are significantly enriched in biological processes such as cellular processes, metabolic processes, localization, response to stimulus, and biological regulation. The molecular functions mainly include binding, catalytic activity, molecule activity, and transporter activity. The cellular components are mainly the cell part, cell, organelle, membrane, and protein-containing complex. In the SAL-6 and SAL-12 groups (Figure 6C), biological processes include cellular process, metabolic process, biological regulation, and regulation of biological process. Molecular functions include catalytic activity, binding, transporter activity, and structural molecule activity. Cellular components include the cell part, cell, membrane, membrane part, organelle, and protein-containing complex. The KEGG enrichment function was employed to analyze the enrichment of the DEPs. In the SAL-0 and SAL-6 groups (Figure 6D), there is evidence of amino sugar and nucleotide sugar metabolism, ribosome, protein processing in endoplasmic reticulum, ribosome biogenesis in eukaryotes, and lysosome. In the SAL-0 and SAL-12 groups (Figure 6E), there is evidence of purine metabolism, biosynthesis of cofactors, amino sugar and nucleotide sugar metabolism, folate biosynthesis, and lysosome. In the SAL-6 and SAL-12 groups (Figure 6F), glutathione metabolism, metabolism of xenobiotics by cytochrome P450, drug metabolism—cytochrome P450, drug metabolism—other enzymes, amino sugar and nucleotide sugar metabolism, and biosynthesis of cofactors are observed. The results of proteomics analyses indicate that metabolic dysfunction and reduced organismal biofactor synthesis and lysosome-related functions were present in *M. rosenbergii* under low-salt stress.

### 2.5. Metabolome Analysis

The objective was to identify differential metabolites (DMs) from the list of major substances in the samples, with a preset *p* < 0.05 and VIP > 1.00. As shown in Figure 7A, SAL-0 was compared to the SAL-6 and SAL-12 groups, and 9152 and 4797 DMs were detected, including 8474 and 2958 upregulated DMs and 678 and 1839 downregulated DMs, respectively. A total of 8510 DMs (620 upregulated DMs and 7890 downregulated DMs) were detected in the SAL-6 and SAL-12 group comparisons. The Venn diagram shows that 354 co-existence DMs were observed in these three groups (Figure 7B). The content of the tyramine, trans-1,2-cyclohexanediol, sorbitol, hydroxy octadecadienoic acid, and uridine diphosphate-N-acetylglucosamine metabolites were significantly downregulated, while the acetylcholine chloride and chloroquine metabolites were significantly upregulated in *M. rosenbergii* under low-salt stress (Figure 7C–E). Furthermore, notable alterations were observed in the concentrations of the stearic acid, vanillic acid, quinic acid, cochineal, acetylcholine, catechol, and NP-007727 metabolites.

### 2.6. Multi-Omics Correlation Analysis

For the combined analysis of the proteome and transcriptome, we first obtained the quantitative detection analysis junction of the proteome and transcriptome. The corresponding proteins and transcripts were then extracted. Then, the differential protein and corresponding differential transcript relationships were elucidated to show the consistency of the relationship pairs before mapping the protein and the associated transcript to the associated metabolic pathway. Finally, the corresponding genes were enriched by GO and KEGG. According to the results of the quantitative and difference analysis of the proteome and transcriptome, Pearson’s correlation coefficient was analyzed and the results were plotted. The transcriptome threshold line was |log2FC| = 1, which means the difference factor was more than two times. The Pearson’s correlation value in SAL-0 vs. SAL-6, SAL-0 vs. SAL-12, and SAL-6 vs. SAL-12 were −0.0117, −0.0111, and 0.01683, respectively (Figure 8A–C). In addition, three, six, and one corresponding differential genes and proteins were obtained in the SAL-0 vs. SAL-6, SAL-0 vs. SAL-12, and SAL-6 vs. SAL-12 group comparisons, respectively (Figure 8D–F). To determine which metabolites and genes are related to each other, a loading plot of different omics was plotted for the associated partial variables (genes or metabolites), with the top 20 variables of each omics shown in red. A multi-omics analysis revealed a strong correlation between two metabolites, chloroquine and 5,6-DHet, with low-salinity stress (Figure 9), which suggests that immune function may be affected.

## 3. Discussion

The higher salinity requirements in the early stages of *M. rosenbergii* resulted in excessive salinity stress in the culture water, which is detrimental to the development of the aquaculture industry [26]. Consequently, low-salt treatment or domestication is imperative, although the precise response and regulatory mechanism remain unclear. The present study was conducted to investigate the responses and effects of acute low-salinity exposure on larvae of *M. rosenbergii*. Initially, we assessed the variations in the biochemical parameters of *M. rosenbergii* under different salinity levels. It was found that low salinity significantly inhibited the GSH-Px, and SOD activities of *M. rosenbergii*. GSH-Px serves as a key enzyme in catalyzing the oxidation of GSH within the glutathione redox cycle. Its primary function is to specifically facilitate the reaction between GSH and ROS, resulting in the formation of GSSG, thereby safeguarding biological membranes from ROS-induced damage and preserving cellular function. Additionally, GSH-Px possesses the capacity to safeguard the liver, enhance the immune system, counteract the deleterious effects of harmful metal ions, and bolster resistance to radiation [27,28,29,30]. In aquaculture, researchers investigated the impact of temperature and salinity on the levels of enzymes such as GSH-PX in the pearl oyster to assess the body’s immunity [31]. SOD is widely found in animals, plants, microorganisms, and cultured cells and catalyzes the formation of H_2_O_2_ and O_2_ from superoxide anions [32]. SOD is an enzyme that scavenges superoxide anions and generates H_2_O_2_, playing a crucial role in the biological antioxidant system [33,34]. These results showed that low salinity levels have a negative impact on the immune system and result in the impairment of auto-oxidative function in larvae of *M. rosenbergii* larvae.

To further uncover the influence of low salinity on *M. rosenbergii* larvae, transcriptomics, proteomics, and metabolomics were used. Transcriptomic analysis revealed that the DEGs were predominantly concentrated in pathways associated with the body’s metabolism, immune response, and inflammatory regulation. This suggests that low-salt stress may result in metabolic disorders and the impairment of immune function in *M. rosenbergii*. Furthermore, the mRNA expression levels of *MIH*, *JHEH*, *EcR*, *Caspase 3* and *Caspase 8* were significantly altered under low-salinity conditions. The relative expression of enzyme genes such as *MIH*, *EcR*, and *JHEH* largely reflects the differences in molting of *M. rosenbergii* under different salinity conditions and is also helpful in determining the tolerance level of *M. rosenbergii* under salinity stress [35]. MIH belongs to the neuropeptide hormone family that is secreted by the X organ–sinus gland complex, and *EcR* is upstream of the *MIH* gene [36,37]. The existence of transcriptional regulatory sites within the promoter region indicates that the expression level of the *MIH* gene is modulated by ecdysone [38]. The relative expression of the *MIH* gene in the hepatopancreas was found to be upregulated with increasing salinity in the present study, which suggests that high salinity factors inhibit the molting behavior in *M. rosenbergii*, as ecdysone is antagonized. During the pre-molting period, the relative expression of the *EcR* gene was elevated, indicating a negative impact on the expression level of the *MIH* gene exerted by the molting hormone receptor. In growth-impaired shrimp, the relative expression of the *EcR* gene was lower than that of normal shrimp, while the relative expression of the *MIH* gene in growth-impaired shrimp was significantly higher than that of normal shrimp [39,40]. These findings indicate that low-salinity stress promoted molting but may have induced growth impairment in *M. rosenbergii*. The role of JHEH in the growth and development of crustaceans is of great importance [41]. The findings indicate that acute salinity exposure adversely affects the growth performance of the *M. rosenbergii* larvae.

At the proteomic level, exposure to low salinity results in the modification of proteins, primarily those involved in metabolic processes, regulation of biofactor synthesis, and cellular processes. The results are comparable to those obtained from transcriptomic analysis. Energy metabolism enzymes are important protein substances in crustaceans that regulate the process of energy metabolism. Their vitality is affected by salinity, and discomfort caused by low salinity greatly reduces their vitality [42]. Sugar metabolism is an important process of energy metabolism in animals, where glucose is firstly glycolyzed into pyruvate to provide energy for the organism [43]. The study found that the processes related to gluconeogenesis would be significantly enhanced in the low-salinity group, further verifying the conclusion of the previous transcriptomics study. It is evident that the rate of glycolysis was accelerated after being stimulated by low salinity, which, in turn, promoted ATP production and NAD+ regeneration. Salinity is a crucial factor that affects enzyme activity, and salinity stress, as well as exogenous stimuli such as light, temperature, and pH, etc., have a significant impact on aquatic animals. These factors can force aquatic animals to regulate their osmotic pressure, which can affect their growth and development by accelerating or slowing down metabolism and enhancing or inhibiting their digestive enzyme activity [44,45].

At the metabolomic level, the levels of tyramine, trans-1,2-cyclohexanediol, and sorbitol were significantly decreased, while the levels of acetylcholine chloride and chloroquine were significantly increased in *M. rosenbergii* under low-salt stress. Tyramine, a neuroactive chemical, has been demonstrated to markedly elevate immune indicators such as lysozymes in *Litopenaeus vannamei* within the hour following tyramine injection [46]. This study provides further support for the hypothesis that low-salt stress induces immune dysfunction in *M. rosenbergii*. In addition to its efficacy as an anti-malarial drug, chloroquine has been shown to exert adverse effects on the immune system, including the inhibition of autophagy and reduction in protein degradation in vivo [47]. Furthermore, notable alterations were observed in the levels of several metabolites, including stearic acid, vanillic acid, quinic acid, cochineal, acetylcholine, and catechol. It has been demonstrated that stearic acid has the capacity to significantly promote mitochondrial fusion and regulate morphological and energy metabolism functions within a period of three hours following intake [48]. Similarly, vanillic acid, a phenolic compound, exerts beneficial effects on mitochondrial function by reducing ROS production, regulating energy metabolism and protecting the heart [49]. This suggests that *M. rosenbergii* modulates its energy metabolism pathways and related energy metabolism enzymes in response to low-salinity stress, and that the metabolites also exhibit corresponding enrichment and inhibition. Furthermore, metabolic studies of low-salinity stress in *Portunus trituberculatus* have demonstrated that organic acids, such as pyruvate and malate, can also affect energy metabolism [50]. The combined results of transcriptomic and proteomic analyses indicate that the metabolite changes induced by low-salt stress may further induce the synthesis of biokines and immune dysfunction, which, in turn, may affect the growth and development of *M. rosenbergii* larvae.

## 4. Materials and Methods

### 4.1. Animals and Experiments Design

*M. rosenbergii* (1.0 ± 0.5 cm) were sourced from Zhejiang Nantaihu Freshwater Aquatic Seed Industry Co., Ltd. (Huzhou, China). *M. rosenbergii* were then cultured in a circulating system and fed twice daily with commercial feed. The circulation system maintained 26 ± 2 °C temperature, 7 ± 1 pH, and >8 mg/L dissolved oxygen. After acclimation for 2 weeks in the lab conditions, 360 *M. rosenbergii* were randomly allocated to three groups (*n* = 40 per group) and exposed to different salinity of 0‰ (SAL-0), 6‰ (SAL-6), and 12‰ (SAL-12) for 120 h, respectively. Each treatment group was repeated three times.

### 4.2. Biochemical Parameters Measurement

After treatment, hepatopancreas and muscle of *M. rosenbergii* were sampled (*n* = 5 per group). Each sample was tested three times. The activities of CAT, SOD, and GSH-Px activities in each group were quantified using kits in accordance with the instructions (Nanjing Jiancheng Bioengineering Institute, Nanjing, China). In brief, 100 mg of tissue sample was taken from each treatment group and prepared as a 10% tissue homogenate on ice by adding a volume of saline nine times the weight of the tissue in a ratio of weight (g)/volume (mL) = 1:9. Subsequently, the sample was centrifuged at 2500 rpm for 10 min at a low temperature, after which the supernatant was collected for analysis. Among the various biochemical parameters analyzed, CAT and GSH-PX were quantified by spectrophotometry at 405 nm and 412 nm, respectively, while SOD was determined by microplate reader at 450 nm.

### 4.3. RNA-Seq Library and Data Analysis

RNA-Seq was performed on the *M. rosenbergii* hepatopancreas from different groups by Nanjing Personal Gene Technology Co., Ltd. (Nanjing, China). Each sample was tested three times. Total RNA of *M. rosenbergii* was prepared from the hepatopancreas using the Trizol method (Invitrogen, Waltham, MA, USA), as previously described. An RNA-seq library was constructed and sequenced according to the same protocol [51]. Raw readings were trimmed and quality-controlled, resulting in the remaining clean reads. Genes were classified as the differentially expressed genes (DEGs) if false discovery rate ≤0.05 and |log2 (fold-change)| ≥ 1.

### 4.4. Detection of Differentially Enriched Genes by qRT-PCR

The *M. rosenbergii* hepatopancreas in each group (*n* = 3) was sampled and tested three times. The total RNA and cDNA were synthesized as previously described [51]. Specific primers are listed in Table 1. The relative gene expression was quantified using the 2^−ΔΔCt^ method, with *β-actin* serving as the internal reference gene. A 15 μL system was established, comprising 7.5 μL of MIX (Vazyme, Nanjing, China), 5.2 μL of deionized water, 0.4 μL of forward primer, 0.4 μL of reverse primer, and 1.5 μL of cDNA. qRT-PCR was performed using CFX96 Real-Time PCR detection System (BioRad, Hercules, CA, USA). The qRT-PCR procedure was provided as below: 95 °C for 5 min, denaturation with 40 cycles at 95 °C for 15 s, followed by annealing and extension at 56 °C for 60 s.

### 4.5. Proteome Sequencing and Analysis

The *M. rosenbergii* hepatopancreas in each group (*n* = 3) was sampled and the proteome of each sample was sequenced and analyzed by Nanjing Personal Gene Technology Co., Ltd. Each sample was tested three times. Mass spectrometry data were obtained from a Triple Pro system. The MaxQuant software (version 1.6.14) was used to filter the data. Proteins were judged to be significantly regulated (*p* ≤ 0.05) for multiplicity changes ≥1.5 (upregulated) or ≤0.67 (downregulated). GO and KEGG enrichment analyses were conducted on all significant DEPs.

### 4.6. Metabolomics

The *M. rosenbergii* hepatopancreas in each group (*n* = 6) was sampled and sent to Nanjing Personal Gene Technology Co., Ltd. for metabolomics analysis. Each sample was tested three times. The samples were treated as previously described [52]. Briefly, the requisite quantity of a precisely weighed sample was introduced to a 2 mL centrifuge tube, and 1000 μL of a tissue extract (75% methanol/chloroform, 25% water) was subsequently added. Grinding was repeated twice, placing 3 steel balls in a tissue grinder and grinding at 50 Hz for 60 s. Next, a room temperature ultrasound was performed for 30 min followed by an ice bath for 30 min. The mixture was centrifuged for 10 min at 12,000 rpm and 4 °C. All supernatant was transferred to a new 2 mL centrifuge tube, concentrated and then dried. The sample was re-dissolved by accurately adding 200 μL of 50% acetonitrile solution (stored at 4 °C) prepared from 2-amino-3-(2-chlorophenyl)-propionic acid (4 ppm). Then, the supernatant was filtered through a 0.22 μm filter and transferred to a detection vial for LC-MS.

The Ropls software (R 4.3.3) was utilized for all multivariate data analyses and modeling. The construction of the models was based on data scaling techniques, including principal component analysis, orthogonal partial least squares discriminant analysis (PLS-DA) and partial least squares discriminant analysis (OPLS-DA). Metabolic profiles can be analyzed as score plots, with each sample represented by a point. The generation of the loading and S-plots provides information about the metabolites that affected the clustering of the samples. All the models evaluated were tested for overfitting using the alignment test method. The descriptive performance of the model was evaluated by the R2X (cumulative) and R2Y (cumulative) values, and perfect model R2X (cum) value of 1 and an R2Y (cum) value of 1. The predictive performance was assessed by Q2 (cum), with a perfect model resulting in a Q2 (cum) value of 1, and the permutation test. The R2 and Q2 values at the *Y*-axis intercept must be lower than Q2 and R2 for the non-placement model, and this type should not be able to be predicted by the replacement model. OPLS-DA permits the utilization of variable projected importance (VIP) to identify differential metabolites and filter out variables contributing to the classification. This is achieved through the use of *p*-values, variable importance projections (VIP) generated by OPLS-DA, and multiplicity of differences (FC). Finally, *p* value < 0.05 and VIP values > 1 were considered to be statistically significant metabolites.

Pathway analysis of differential metabolites by MetaboAnalyst 4.0 combines powerful pathway enrichment analysis results with pathway topology analysis results [53]. Subsequently, in order to achieve a higher level of biological interpretation of system function, the identified metabolites from the metabolomics analysis were mapped in the KEGG pathway and the resulting pathways and metabolites were visualized in the KEGG Mapper tool [54].

### 4.7. Statiscical Analyses

All data results were statistically analyzed using IBM SPSS Statistics 26.0 software (SPSS Inc., Chicago, IL, USA) and plotted using Prism GraphPad 9.0 (GraphPad, San Diego, CA, USA). The enzyme activity levels in the hepatopancreas and muscle, and gene mRNA expression levels, in the hepatopancreas of *M. rosenbergii* after the salinity treatments were expressed as mean ± standard deviation (SD) and analyzed in post hoc comparisons by one-way analysis of variance (ANOVA) and Tukey’s analysis.

## 5. Conclusions

In this study, the effects of low-salt stress on biological functions such as molting and immune response, as well as the organismal metabolites in the larvae of *M. rosenbergii* were investigated using multi-omics combined with histological analysis. These findings provide new perspectives on the molecular mechanisms of the effects of low salinity on the larvae of *M. rosenbergii*, provide crucial theoretical support for low-salinity juvenile prawn cultivation and acclimation, and provide a scientific foundation for addressing issues in the development of the *M. rosenbergii* industry, which has significant academic and practical implications.

## Figures and Tables

**Figure 1 ijms-25-06809-f001:**
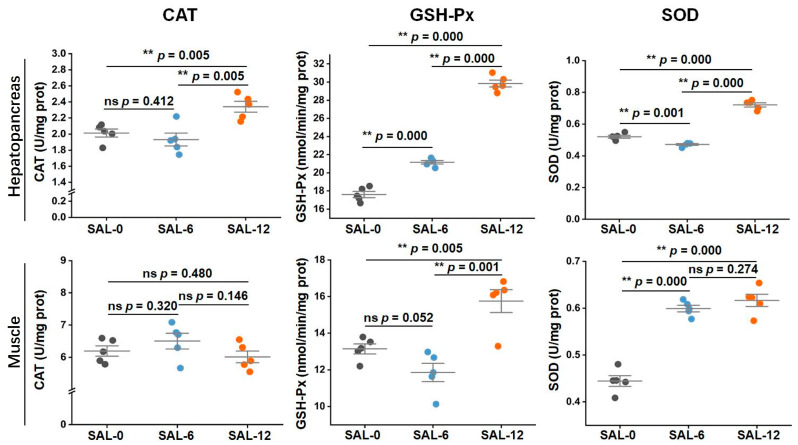
Following a 120 h salt treatment with varying concentrations of *M. rosenbergii*, samples of the hepatopancreas and muscle were collected. The CAT, SOD, and GSH-Px were then assayed according to the protocol described in the kit. Data are expressed as mean ± standard deviation (SD). ** *p* < 0.01, ns *p* > 0.05.

**Figure 2 ijms-25-06809-f002:**
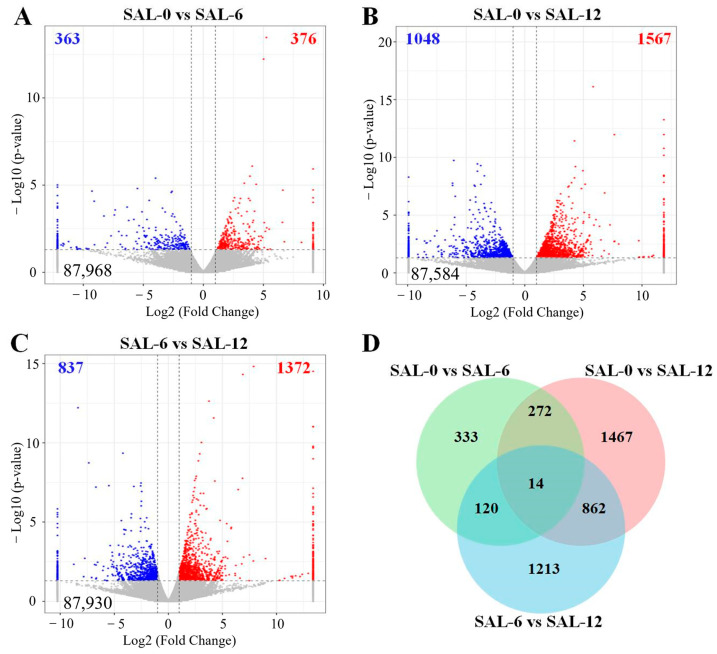
A transcriptome analysis was conducted on DEGs in *M. rosenbergii* under varying salinity exposures. (**A**) Volcano plot of DEGs in SAL-0 vs. SAL-6 group; (**B**) Volcano plot of DEGs in SAL-0 vs. SAL-12 group; (**C**) Volcano plot of DEGs in SAL-6 vs. SAL-12 group. Red-colored plots illustrate up-regulated genes, blue-colored plots illustrate down-regulated genes and gray plots reflect genes that did not show changes in expression; (**D**) Venn figure shown the number of DEGs between different groups.

**Figure 3 ijms-25-06809-f003:**
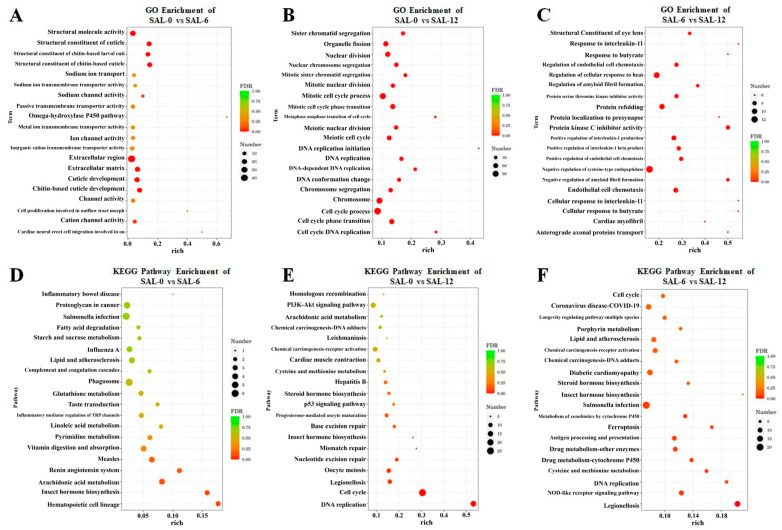
KEGG and GO enrichment analyses of DEGs from *M. rosenbergii* at different salinities. (**A**) GO enrichment in transcriptome analysis of DEGs in the SAL-0 and SAL-6 groups; (**B**) GO enrichment in transcriptome analysis of DEGs in the SAL-0 and SAL-12 groups; (**C**) GO enrichment in transcriptome analysis of DEGs in the SAL-6 and SAL-12 groups; (**D**) KEGG enrichment in transcriptome analysis of DEGs in the SAL-0 and SAL-6 groups; (**E**) KEGG enrichment in transcriptome analysis of DEGs in the SAL-0 and SAL-12 groups; (**F**) KEGG enrichment in transcriptome analysis of DEGs in the SAL-6 and SAL-12 groups.

**Figure 4 ijms-25-06809-f004:**
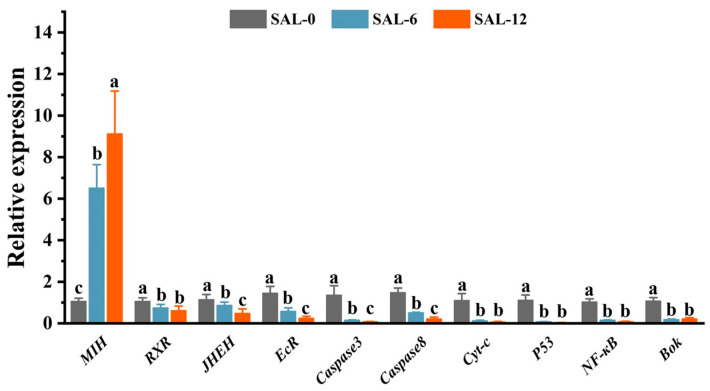
The mRNA expression levels of the genes in the hepatopancreas of *M. rosenbergii* were quantified using qPCR following the treatment of the hepatopancreas with different concentrations of salt for 120 h. The relative gene expression was quantified using the 2^−ΔΔCt^ method, with *β-actin* serving as the internal reference gene. Data were expressed as mean ± standard deviation (SD) and analyzed in post hoc comparisons by one-way analysis of variance (ANOVA) and Tukey’s analysis. Data at the same sampling time with different lowercase letters (a, b, and c) are significantly different (*p* < 0.05).

**Figure 5 ijms-25-06809-f005:**
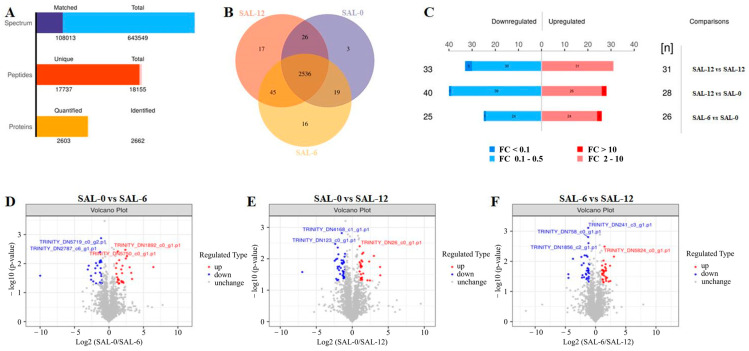
DEPs in *M. rosenbergii* hepatopancreas. (**A**) Statistical histogram of identification and quantitative results; (**B**) Venn diagram of all reidentified proteins; (**C**) DEPs in different groups; (**D**) Volcano plot of DEPs in the SAL-0 vs. SAL-6 groups, (**E**) SAL-0 vs. SAL-12 groups, (**F**) SAL-6 vs. SAL-12 groups. Red-colored plots illustrate upregulated genes, blue-colored plots illustrate downregulated genes and gray plots reflect genes that did not show changes in expression.

**Figure 6 ijms-25-06809-f006:**
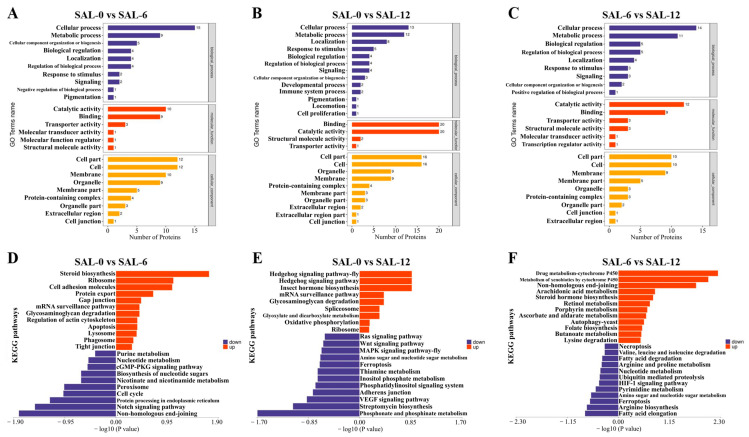
GO and KEGG enrichment analysis of DEPs in the hepatopancreas of *M. rosenbergii*. (**A**) GO enrichment in transcriptome analysis of DEPs in the SAL-0 and SAL-6 groups; (**B**) GO enrichment in transcriptome analysis of DEPs in the SAL-0 and SAL-12 groups; (**C**) GO enrichment in transcriptome analysis of DEPs in the SAL-6 and SAL-12 groups; (**D**) KEGG enrichment in transcriptome analysis of DEPs in the SAL-0 and SAL-6 groups; (**E**) KEGG enrichment in transcriptome analysis of DEPs in the SAL-0 and SAL-12 groups; (**F**) KEGG enrichment in transcriptome analysis of DEPs in the SAL-6 and SAL-12 groups.

**Figure 7 ijms-25-06809-f007:**
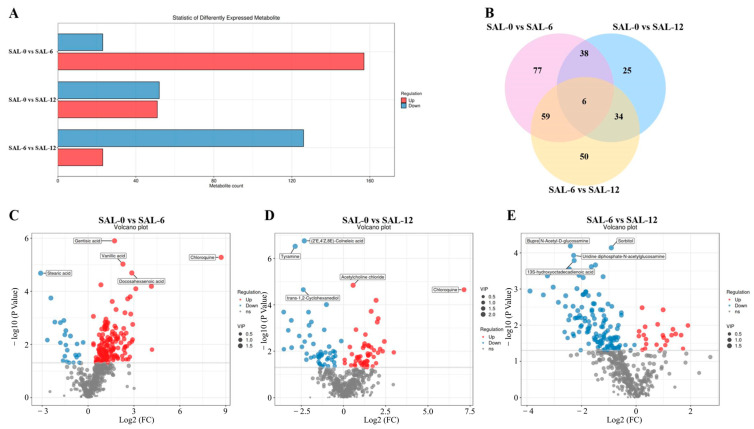
Differences in metabolites (DM) in different groups. (**A**) Differences in the amount of DM in different salinity groups; (**B**) Venn diagrams of all recharacterized proteins; (**C**) Volcano plot of DM in SAL-0 vs. SAL-6 groups; (**D**) Volcano plots of DM in the SAL-0 vs. SAL-12 group; (**E**) Volcano plot of DM in the SAL-6 vs. SAL-12 group. Red plots illustrate upregulated genes, blue-colored plots illustrate downregulated genes and gray plots reflect genes that did not show changes in expression.

**Figure 8 ijms-25-06809-f008:**
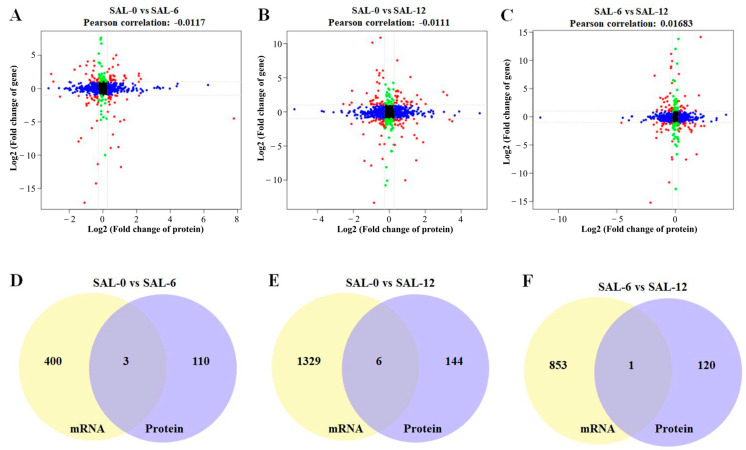
Co-analysis of the transcriptome and proteome. (**A**) Pearson’s correlation analysis of SAL-0 vs. SAL-6 groups; (**B**) Pearson’s correlation analysis of SAL-0 vs. SAL-12 groups; (**C**) Pearson’s correlation analysis of SAL-6 vs. SAL-12 groups; (**D**) Venn diagram of corresponding differential proteins and corresponding differential transcripts of the SAL-0 vs. SAL-6 groups; (**E**) Venn diagram of corresponding differential proteins and corresponding differential transcripts of the SAL-0 vs. SAL-12 groups; (**F**) Venn diagram of corresponding differential proteins and corresponding differential transcripts of the SAL-6 vs. SAL-12 groups.

**Figure 9 ijms-25-06809-f009:**
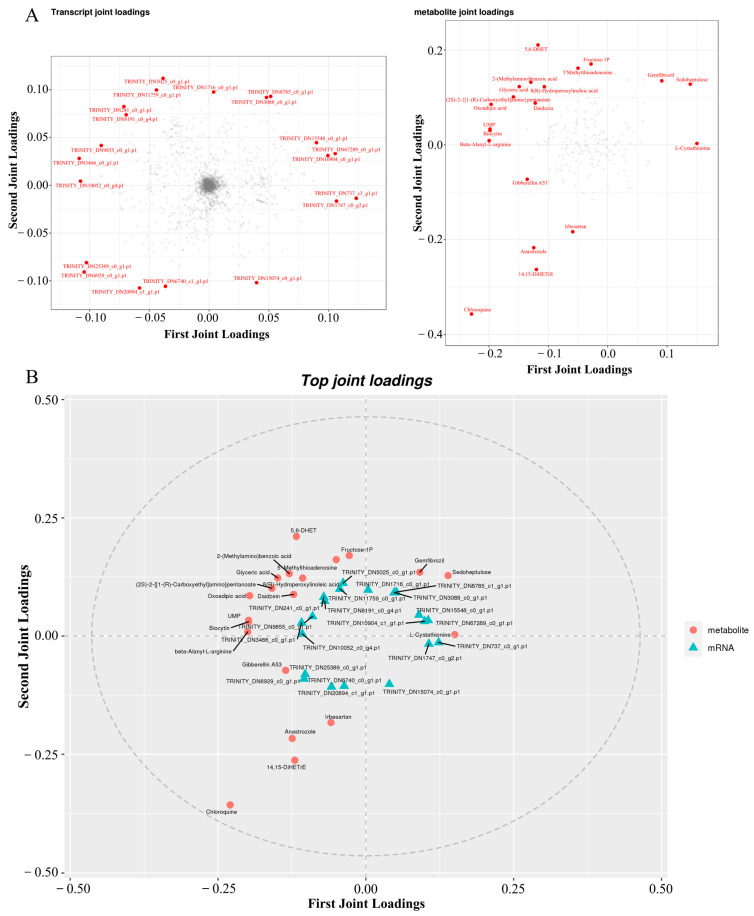
Co-analysis of transcriptome, proteome, and metabolism analysis. (**A**) O2PLS loading diagram; (**B**) Correlated load diagram.

**Table 1 ijms-25-06809-t001:** Primers used for qPCR analysis.

Gene	Primers
*β-actin*	F: CGACGGTCAGGTCATCACCA
	R: ACGTCGCACTTCATGATGGA
*MIH*	F: CCAGACAACGCAAGGGATCT
	R: TCGTCGCATACCCTGACAAC
*RXR*	F: GCGAGAAGCGGTCCAGGAGG
	R: GGTGGGGTCTGAGTTGAGTTCTGC
*JHEH*	F: CTTCCTGAGAGCAAGTGCCAAA
	R: AGGCTTCGTCAACAATGGCAAA
*EcR*	F: AAGAGCCGCATAAAGTGGAGAAGC
	R: AGGTCGGTCAGGATGTTCAGGAG
*Caspase3*	F: CGGATTCAAACGCGATGACC
	R: GACGACAACGTGGTCTGACT
*Caspase8*	F: GCGAAAGAACTACTCGGCCG
	R: AGCAGCAGCCAGGAACTTGT
*Cyt-c*	F: TGGGTGACGTAGAAAAGGGC
	R: TGCCTTGTTAGCGTCAGTGT
*P53*	F: CCCTCGTCATCAGTTGCCAG
	R: TGAAGGAGTTGCTGGGGTTAC
*NF-κB*	F: AGATGCCGAGGAGGTATGGA
	R: GCGTCGTTGAAATGCGATGT
*Bok*	F: TCAGTACTTCAAATGCTAGTGCTG
	R: CGTCATAAACCGTCCCTA

## Data Availability

All data are available in the main text.

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
