# Peer review of "Regulation and Response Mechanism of Acute Low-Salinity Stress during Larval Stages in Macrobrachium rosenbergii Based on Multi-Omics Analysis"

_ijms, 2024, doi:10.3390/ijms25126809_

Round 1

Reviewer 1 Report

Comments and Suggestions for Authors

This study reports about transcriptomics, proteomics and metabolomics on the possible role and mechanism of low salinity exposure on larvae of Macrobrachium rosenbergii. In general the Figure legends contain too little text so that a reader cannot study a Figure without going back to M&M and Results to hopefully get to understand what the Figure shows! This manuscript is of interest but is in need of a very major revision before being considered for inclusion in any journal.

In M& M it is stated that carapace and muscle were isolated for further studies. For this reviewer carapace is very strange. This is not a tissue but rather a portion of the body with several tissues,  so what is collected?

It says that each sample was tested three times! This must mean technical replicates were done and it should be biological replicates that is tissue or tissue sample from three individual shrimp or groups of shrimp. What do the authors mean?

Enzyeme activities of SOD,GSH and  CAT were tested and this reviewer wonders if proper kinetic conditions were present during assay of enzyme activities? Please specify!

RNA seq. on carapace was made? On one animal or? Again what is carapace in this sample? And why was ”carapace ” used? Please provide arguments for this choice?

It is stated that “sample was sequenced and analyzed by Nanjing Personal Gene Technology Co., Ltd. Eachsample was tested three times”  Again this means technnical replicates or? It should be biological replicates which are tested I hope. 

In Figure 1 it seems strange that at SAL-6 GSH is the same as control in muscles whereas at SAL-12 very much higher in muscles? Is this true and correct?  SOD is higher than control at both salt conditions in muscles but not in carapace !Why is this difference since in carapace there are muscles as well or?  

Is not the activity of at least SOD very low? Please give detailed information so it possible for a reader to make proper calculation of true enzyme activity for example absorbance where appropriate!

”In SAL-0 vs SAL-12 group the DEGs mainly enriched in hematopoietic cell 124 lineage, sister chromatid segregation, organelle fission, nuclear division, nuclear chromo- 125 some segregation, mitotic sister chromatid segregation (Figure 2G” This reviewer is unable to see what genes are affected because of the small size of the text and it should be interesting if the authors can detail the RNAs involved in hematopoeisis in this shrimp since very little is known about this? Please specify!

The validation by qPCR was this done on tissue from three different individual animals? Please detail!

Figure 4 and 5 too small to read! Figure 6 can also be made larger.

The metabolomic data mainly show that secondary metabolites seems to be affected and the authors seem to not discuss in some detail what the results of the metabolomic results might indicate?

The results of the validation data to this reviewer seems that the most interesting result is that the molting inhibiting hormone (MIH) was by far the most effected mRNA. All other mRNAs do not seem to change in a statistically significant way. So this obviously means that molting might be inhibited at quite low salt concentrations  and this would at least make this reviewer more interested in focusing more research on this finding. If animals are exposed to low salt stress and the mRNA for MIH is silenced can then the animals start to molt for example! Is this effect reversible that is if 120 hrs exposure to low salt then how long time will it take to restore the mRNA level of MIH and so on. This could also mean a lot for the farming of this freshwater shrimp so these studies may be of great interest.

Comments on the Quality of English Language

Edit English

Reviewer 2 Report

Comments and Suggestions for Authors

The manuscript by Li, et al., describes a study using multi-omics, enzymatic and qPCR approaches to study the effects of acute salinity transfers of larval Macrobrachium rosenbergii.

General Comments:

Overall, this is an interesting study and advances our knowledge regarding the effects of acute salinity transfer to this important aquaculture species.  However, I have a number of major concerns that need to be addressed before the manuscript is ready for publication.  Perhaps most importantly, there is virtually no description of the statistical approaches used in the study.  In the one case where the statistics are described, they seem inappropriate.  Also, the justification for the study is not clearly presented. Why is salinity a concern for culturing these prawns?  How does this relate to their catadromous life history?  While the authors briefly touch on this in the final paragraph of the Introduction, a clear impetus for the study is not provided. Moreover, the authors do not explain why they selected carapace and muscle tissue to focus on for their analyses.  Many of the methods lack sufficient detail as well.  

Specific Comments:

Title:  suggest adding life stage

Line 12:  here and throughout “sapling” is inappropriate. also, unclear what is meant by water pollution in this context; perhaps stress would be a better choice (also in Line 42)?

Line 14:  biomarkers of what? salinity exposure or salinity effects?

Line 24:  chlorquine or chloroquine?  is this the correct term?  a google search identifies this as an antimalarial drug

Line 35:  this is a prawn rather than a shrimp

Lines 52-54:  sentence is poorly constructed and hard to follow

Line 57:  what is meant by “salinity safety value”?

Lines 74-76:  unclear what the link is between histological studies and single-omics studies in this context

Line 86:  “mechanism of exposure” is inappropriate – this should be a mechanism of the animal’s response or tolerance

Line 93 and throughout:  please spell out abbreviations on first use

Figure 1:  the statistics here are questionable.  given more than two salinities are tested, an ANOVA followed by post-hoc tests would be more appropriate

Figure 2:  the figure text is far too small to read; also, suggest splitting into two figures with A-D in one and E-J in another

Figure 3:  statistical approach is not included; also what is the data expressed relative to?  typically, the lowest level of expression is set to 1 and all others expressed relative to that.  if you’re not comparing across transcripts, then the lowest level of expression for each gene should be set to 1

Figures 4 and 7:  again, text is far too small

Line 277:  contamination with what?  this seems like an odd statement.  salt in this context does not seem to be a contaminant per se, so I think a different word or phrasing would be more appropriate.

Line 278:  domestication?

Line 280:  but your analyses were not conducted in vivo, please revise

Line 293:  the authors do not mention immune related effects in the paragraph to support this statement

Line 309:  hepatopancreas?  this is the first mention of this tissue

Section 4:  please add a section for statistical analyses

Lines 361 and 443:  seedlings is inappropriate

Line 364:  life stage?

Line 370:  so is this n=40 for each replicate within a treatment?

Section 4.2:  please add more detail. for example, are these measured using a colorimetric assay?  what was the equipment used to make the measurements?  what was the temperature of the assays and was temperature controlled?

Section 4.4:  more details are needed here.  what are the actual reaction conditions; i.e., volumes and primer concentrations, amount of cDNA, etc. of reaction components? was the RNA treated with DNAse to eliminate genomic contamination?  Was the cDNA diluted prior to use?  The 2 delta delta approach assumes all reaction efficiencies are close to 100% - did the authors confirm this?  Were the amplicons sequence verified?  Was a melt analysis carried out after each reaction to confirm specificity?  Were negative controls included?  Was B-action confirmed not to change with treatment?  also, the thermal cycling parameters seem odd as the authors only list a very long annealing temperature (60 s), but no extension phase – is 56 C supposed to serve for both?  again, this seems odd and atypical.

Comments on the Quality of English Language

Much of the manuscript suffers from poor grammar and would greatly benefit from editing by a native English speaker. 

Round 2

Reviewer 1 Report

Comments and Suggestions for Authors

This manuscript has been revised to accommodate most of my criticism. The only part which remains not properly answered is the use of carapace and the authors say they used the shell and then "shell" does not contain any tissues except maybe remnants of epithelia and attached muscle. So this reviewer still needs to know what the "shell"  carapace really contains and this reviewer guess that most scientists working with crustaceans need to know this as well. So once again the "carapace " is not a tissue but rather the cuticle with contains proteins,lipds and chitin and no cells except attached cells or epithelia.

Comments on the Quality of English Language

Is this really necessary to fill in with text every time?
